# Data-driven discovery of mechanisms underlying present and near-future precipitation changes and variability in Brazil

Márcia Talita A. Marques[1], Maria Luiza Kovalski[1], Gabriel M. P. Perez[1], Thomas C. M. Martin[1], Edson L. S. Y. Barbosa[1], Pedro Augusto S. M. Ribeiro[1], Roilan H. Valdes[2]

[1]MeteoIA, São Paulo, Brazil
[2]Engie Brazil, São Paulo, Brazil

*Correspondence to*: Gabriel M. P. Perez (gabriel@meteoia.com)

**Abstract.** Untangling the complex network of physical processes driving regional precipitation regimes in the present (1979-2014) and near-future climates (2020-2050) is fundamental to support a more robust scientific basis for decision making in the water-energy-food nexus. We propose a data-driven mechanistic approach to: (Goal 1) identify changes and variability of the regional precipitation mechanisms and (Goal 2) reduce the ensemble spread of future projections by weighting and filtering models that satisfactorily represent these drivers in present climate. Goal 1 is achieved by applying the Partial Least Squares (PLS) technique, a two-sided variant of principal component analysis (PCA), on a reanalysis dataset and 30 simulations of the future climate submitted to CMIP6 to discover the links between global sea-surface temperature (SST) and precipitation in Brazil. Goal 2 is achieved by selecting and weighting the future climate simulations from climate models that better represent the dominant modes discovered by the PLS in the present climate; with this subset of climate simulation, we produce precipitation change maps following IPCC's WG1 methodology. The main mechanistic link discovered by the technique is that the generalised warming of the oceans promotes a suppression of precipitation in Northeast and Southeast Brazil, possibly mediated by the intensification of the Hadley circulation. We show that this pattern of precipitation suppression is stronger in the near-future precipitation change maps produced using our methodology. This demonstrates that a reduction of epistemic uncertainty is achieved after we select models that skillfully represent these mechanisms in the present climate. Therefore, the approach is capable of supporting both a quantitative analysis of regional changes as well as the construction of storylines supported by mechanistic evidence.

## 1 Introduction

Information about near-future regional precipitation change is crucial for planning and managing critical infrastructure, such as hydropower plants, water reservoirs, and city planning. Unpreparedness for changes and variations in regional precipitation regimes may lead to disruption in the water-food-energy supply chains as well as avoidable deaths and damages by flooding and landslides. Although there is a degree of certainty about global precipitation changes (Shepherd et al., 2018),

such as the intensification of the hydrological cycle, a current major challenge in climate change science is informing
planners and decision-makers about regional changes within the critical time-frame of the next three decades.
Within this time frame, the two main sources of uncertainty in regional precipitation changes are model uncertainty and
internal variability (Hawkins and Sutton, 2011). Uncertainty due to the internal variability of the climate system is
impossible to reduce and is aleatoric and related to the chaotic nature of the system (Shepherd, 2019). Model uncertainty, on
the other hand, is epistemic in nature and stems from our limited knowledge of Earth's climate system and from the
challenges in translating this system into computer models. Currently, there are 131 available models on the CMIP6
database, each representing Earth's climate with a range of parameterizations and numerical modelling strategies. From the
full set of models, 29 are available for the relevant experiments for this study.
In this study, we seek for a reduction of the epistemic uncertainty of regional precipitation changes in Brazil through a
data-driven process-based methodology of model selection and weighting. The method discovers the relationships between
sea surface temperature and precipitation in Brazil and evaluates the capability of CMIP6 models to reproduce these
precipitation mechanisms in the present climate. Later, the best models are selected and weighted to produce refined
precipitation maps. Due to the process-based nature of the method, it is also possible to isolate mechanisms and draw
storylines of plausible futures. The paper answers the following questions:
● What are the spatiotemporal links between global sea-surface temperature (SST) and regional precipitation change
and variability in Brazil? Many patterns have been identified in the literature (Grimm et al., 2000; Coelho et al.,
2002), but here we choose to use a supervised ML approach to systematically identify and quantify their importance
● Can we take advantage of these mechanisms to filter CMIP6 simulations and reduce the epistemic uncertainty of
regional precip changes?
●  What are the projections for precipitation in Brazil over the next 30 years based on a mechanistic filtering and
weighting procedure?

## 2 Materials & Methods

### 2.1 Data-driven discovery of precipitation mechanisms

To discover the underlying mechanisms linking the SST spatiotemporal variability and regional precipitation in Brazil we
employ a data-driven dimensionality reduction method known as Partial Least Squares (PLS) adapted to a latitude-longitude
grid; which has been recently shown to successfully identify circulation mechanisms leading to precipitation (Perez et al,
56 2022).

The PLS method identifies pairs of latent variable vectors $\xi$ and $\omega$ that maximises the information present in $X^t Y$. This
means finding latents variables that represent the maximum covariance between X and Y, where X and Y represent two
arrays of SST and precipitation, respectively; rows of X and Y represent the monthly averaged temporal samples while the

columns represent the spatial lat-lon grid points. The more familiar Principal Component Analysis (PCA) can be seen as a special case where X=Y. The initial set, or mode, of latent variables is determined through the following covariance Eq. (1):

$$Cov(\xi_1, \omega_1) = max_{\|u\|=\|v\|=1} Cov(Xu, Yv) , \tag{1}$$

where u and v are temporally invariant arrays of loadings; in contrast to PCA, PLS yields a pair of loading matrices per component rather than a single loading matrix; the first pair of loading matrices is the one in which the corresponding latent vectors $\xi$ and $\omega$ are the most correlated. The following modes are found through repeating the process on the residuals of each preceding pair.

The interpretation of PLS results should always consider scores and loadings concurrently. A positive loading correlation, coupled with a positive trend in the scores, indicates an increase in signal strength over time. Conversely, when loadings exhibit the same signal but are associated with a negative trend in scores, this suggests a decrease in signal intensity. When evaluating the relationship between two loadings, we first observe how the loading patterns are linked through the score time series. This connection allows us to infer the response of the Y pattern from the X pattern. For instance, if the SST signal is negative at a particular location while the score is positive, this indicates a negative association with the corresponding precipitation loading pattern. A detailed explanation of the method can be found in Wegelin (2000).

## 2.2 Present and future climate datasets

The PLS method was applied to two kinds of climate datasets: firstly, to present climate data from AMIP experiments and reanalysis and, secondly, to the future climate simulations. In the AMIP experiments, atmospheric models are driven by prescribed sea surface temperatures, while radiative forcings and land use are kept constant. This approach helps identify model errors that arise from interactions within the atmosphere, the land surface, or between these components (Eyring et al., 2016). The subsections below describe the methodologies and data behind the present and future climate results.

## 2.2.1 Present climate (AMIP)

The first step was to establish a transfer function linking SST and precipitation month-to-month co-variability using the PLS technique, for the reanalysis and atmosphere-only experiments. The goal is to identify models that accurately represent the transfer function identified in the reanalysis in the present climate. To achieve this, we employ precipitation data derived from the ERA5 reanalysis (Hersbach and Dee, 2016), in addition to precipitation data from 29 AMIP models from the Coupled Model Intercomparison Project Phase 6 (CMIP6), as outlined in Table 1. Before the PLS technique was employed, the ERA5 precipitation data underwent systematic error correction using observations from the Global Precipitation Climatology Project (GPCP, Adler et al., 2018) as a reference through the quantile mapping method, which adjusts probability distributions by individually matching each quantile to the respective quantile of the reference dataset (Jakob et al., 2011). GPCP data has a shorter time period (from 1979) when compared to ERA5 data, which has been available since 1950. Since we aimed to investigate interannual/interdecadal variability and climate change, we chose the longer dataset.

Each precipitation dataset was conservatively gridded to a regular 1°x1° lat-lon grid in a monthly temporal resolution
between 1979 and 2014. SST data was obtained from the COBE dataset, produced by the Japan Meteorological Agency
(Hiragana et al., 2014) , which has long temporal availability and is observations-based.
Table 1 - CMIP6 simulations, their native resolutions, vertical levels and source institutions

| Model | Horizontal resolution | Vertical levels | Variant label | Institution |
| --- | --- | --- | --- | --- |
| ACCESS-CM2 | 1.875° × 1.25° | 85 | r1i1p1f1 | CSIRO |
| ACCESS-ESM1-5 | 1.875 ° x 1.25° | 38 | r1i1p1f1 | CSIRO |
| BCC-CSM2-MR | 2.81° x 2.81° | 46 | r1i1p1f1 | BCC |
| CAMS-CSM1-0 | 1° x 1° | 31 | r1i1p1f1 | CAMS |
| CanESM5 | 2.81° x 2.81° | 49 | r1i1p1f1 | CCCma |
| CESM2-WACCM | 0.9° x 1.25° | 70 | r1i1p1f1 | NCAR |
| CIESM | 1° x 1° | 30 | r1i1p1f1 | THU |
| CMCC-CM2-SR5 | 1° x 1° | 30 | r1i1p1f1 | CMCC |
| CNRM-CM6-1 | 1.4° x 1.4° | 91 | r1i1p1f2 | CNRM-CERFACS |
| CNRM-CM6-1-HR | 1.4° x 1.4° | 91 | r1i1p1f2 | CNRM-CERFACS |
| CNRM-ESM2-1 | 1.4° x 1.4° | 91 | r1i1p1f2 | CNRM-CERFACS |
| EC-Earth3-CC | 0.7° x 0.7° | 91 | r1i1p1f1 | EC-Earth-Consortium |
| EC-Earth3-Veg | 0.7° x 0.7° | 91 | r1i1p1f1 | EC-Earth-Consortium |
| EC-Earth3-Veg-LR | 1.1° x 1.1° | 62 | r1i1p1f1 | EC-Earth-Consortium |
| FGOALS-f3-L | 1° x 1° | 32 | r1i1p1f1 | IAP/CAS |
| FGOALS-g3 | 2° x 2° | 26 | r1i1p1f1 | IAP/CAS |
| GFDL-CM4 | 1° x 1° | 33 | r1i1p1f1 | NOAA-GFDL |
| GFDL-ESM4 | 1° x 1° | 49 | r1i1p1f1 | NOAA-GFDL |
| IITM-ESM | 2° x 2° | 64 | r1i1p1f1 | CCCR-IITM |
| INM-CM4-8 | 2° x 1.5° | 21 | r1i1p1f1 | INM |
| INM-CM5-0 | 2° x 1.5° | 73 | r1i1p1f1 | INM |
| IPSL-CM6A-LR | 2.5° x 1.3° | 79 | r1i1p1f1 | IPSL |
| KACE-1-0-G | 1.9° x 1.3° | 85 | r1i1p1f1 | NIMS-KMA |
| MIROC6 | 1.4° x 1.4° | 81 | r1i1p1f1 | MIROC |
| MPI-ESM1-2-HR | 0.93° x 0.93° | 95 | r1i1p1f1 | MPI-M |
| MPI-ESM1-2-LR | 1.9° x 1.9° | 47 | r1i1p1f1 | MPI-M |
| MRI-ESM2-0 | 1.125° x 1.125° | 80 | r1i1p1f1 | MRI |

| Model | Horizontal resolution | Vertical levels | Variant label | Institution |
|---|---|---|---|---|
| ACCESS-CM2 | 1.875° × 1.25° | 85 | r1i1p1f1 | CSIRO |
| NESM3 | 1.9° x 1.9° | 47 | r1i1p1f1 | NUIST |
| NorESM2-LM | 2° x 2° | 32 | r1i1p1f1 | NCC |
| TaiESM1 | 1.25° x 0.9° | 30 | r1i1p1f1 | AS-RCEC |


The models listed above, through their computational representations of the atmosphere, choices of parameterisation, vertical
levels etc, provide unique numerical representations of the physical climate system. Each of these representations have a
distinct level of skill in simulating the mechanisms of precipitation variability and changes in Brazil.
Therefore, we rank and select the models with higher performance to represent the  SST-precipitation transfer function
revealed by the PLS analysis. This ranking is based on the Normalised Root Mean Square Error (NRMSE), which is
obtained by comparing the PLS scores and loadings, between each model and those derived from the ERA5
reanalysis.Specifically, the RMSE is computed between the scores and loadings from each model and the corresponding
scores and loading from ERA5. Then, these RMSEs are collectively normalized, yielding the NRMSEs, to ensure consistent
scaling across all comparisons. Models with NRMSE values below 0.6 in at least two of the first four PLS components are
then selected. This threshold was chosen so that a representative number of models is kept in the ensemble:  15 models
compose the selected subset. A more strict NRMSE threshold would lead to a substantially smaller subset and reduce the
statistical robustness of the results while a more relaxed threshold would be less effective to reduce the epistemic uncertainty
by keeping models that poorly represent the desired mechanisms. These selected models are singled out as more reliably
representing mechanisms that cause the precipitation in Brazil while the rest is discarded for the remaining analysis.
After the model ranking and selection step, we provide a set of weights that will be later used for model averaging. This set
of weights is found by multiplying the inverse of the NRMSE by the importance of each PLS component; this is done so that
models that perform well in representing more relevant mechanisms are favoured during the model pooling step. The
importance of each PLS component is quantified by the coefficient of determination ($r^2$) of the reconstructed precipitation
using only that component and the original ERA5 precipitation.
**2.2.2 Future climate**
We employ the same PLS methodology on future climate simulations under the SSP2-2.45 scenario between 2020 and 2050;
in this near-future temporal range, we do not expect the choice of scenario to influence the results because scenario
uncertainty in regional precipitation changes only becomes relevant in later decades (Hawkins and Sutton, 2011).
Finally, the effectiveness of this methodology in reducing the  uncertainty of near-future precipitation changes in the CMIP6
ensemble is assessed by comparing the uncertainty of all CMIP6 models listed in Table 1 with the uncertainty of the subset
of models selected by our methodology. The climate change signal was computed for each grid cell by calculating the ratio
(in %) between the anomaly of the ensemble mean climatologies of the SSP2-4.5 scenario for the years 2020-2050 and the
historical period of 1979-2014, divided by the historical. To assess the robustness of the models, we apply the procedure
adopted by the Intergovernmental Panel on Climate Change (IPCC), as outlined in its Sixth Assessment Report, made
available through the Interactive Atlas developed by Working Group I (WGI). This approach determines the robustness of
climate change signals based on a strong model consensus, highlighting where at least 80% of the models agree on the sign
of the predicted changes.

## 128 3 Results and discussion

In this section, we present the results of the analysis for the present and future climates, discussing the underlying
precipitation mechanisms in reanalysis and model data. We also discuss the reduction of epistemic uncertainty of regional
precipitation changes obtained through the selection of models that skillfully represent precipitation mechanisms in the
present climate. In all PLS analysis, the Legal Amazon area was cropped off; this is because precipitation in the Amazon
region presents significantly higher variability in magnitude, dominating the results and washing out patterns in other areas
that are also socioeconomically relevant.

### 135 3.1 Precipitation mechanisms in the present climate (1979-2014)

In the present climate, the first PLS loadings matrix of the SST reveals a prominent positive pattern in the central Pacific
Ocean that aligns with the region dominated by the El Niño/Southern Oscillation (ENSO) phenomenon (Fig. 1a). This
ENSO-like pattern with high statistical significance (unhatched area) extends from the west coast of South America to the
Maritime Continent in the equatorial region, surrounded by a pattern of opposite signal. There is a negative trend in Fig. 1c
scores, indicating a change of sign in the patterns of Fig. 1a. This suggests that in the first half of the timeseries, El Niño
conditions were dominant, while in the second half of the time series, La Niña conditions were. The associated PLS loadings
matrix for precipitation shows a significant positive correlation in South Brazil and a negative correlation in Northeast Brazil
(Fig. 1b). The time series of the associated scores do not show a strong linear trend, reinforcing that this PLS mode is more
associated with a natural variability mechanism like ENSO than to climate change (Fig. 1d).
The global warming trend can explain the mostly positive SST loadings matrix and the increasingly positive scores time
series of the second PLS component (Fig. 2a,c). This warming oceanic pattern is linked to a precipitation reduction in most
of Southeast and Northeast Brazil (Fig. 2b,d). A possible explanation for this precipitation suppression is the expansion of
the Hadley cell under climate change (Lu et al., 2007; Grise & Davis, 2020) and, consequently, the restriction of the
equatorward motion of extratropical cyclones and their fronts, which are important precipitation mechanisms in Southeast
Brazil (Perez et al., 2021). Perez et al. (2022) has shown that a temporary intensification of the Hadley circulation during
positive NAO events leads to precipitation suppression in Southeast Brazil.

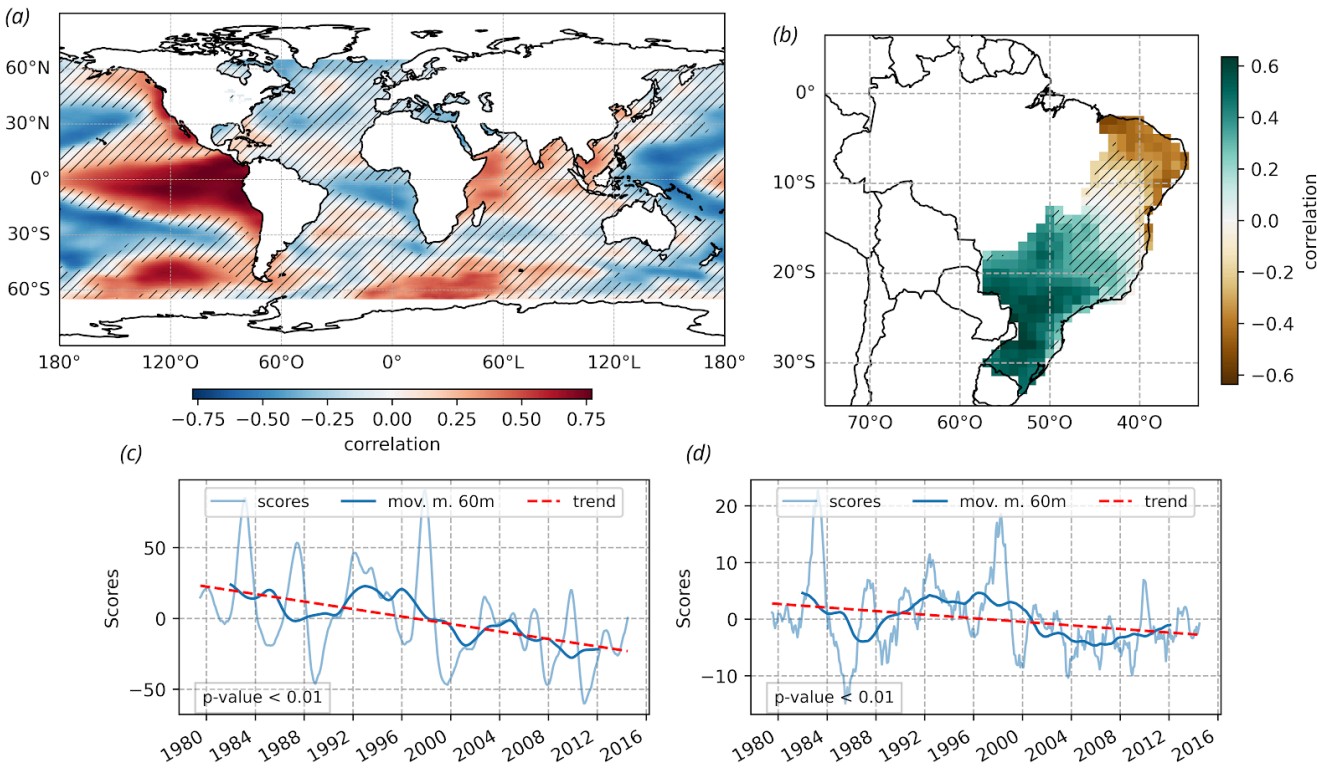


**Figure 1: First component of the PLS methodology applied using monthly precipitation data from ERA5 and SST data from COBE between 1979 and 2014. (a, b) The spatial maps represent the loadings matrices, where the hatchings represent areas where the statistical confidence on the sign of the anomaly is lower than 95%. (c, d) The time series represents the scores and the p-values indicate the statistical significance of the results.**

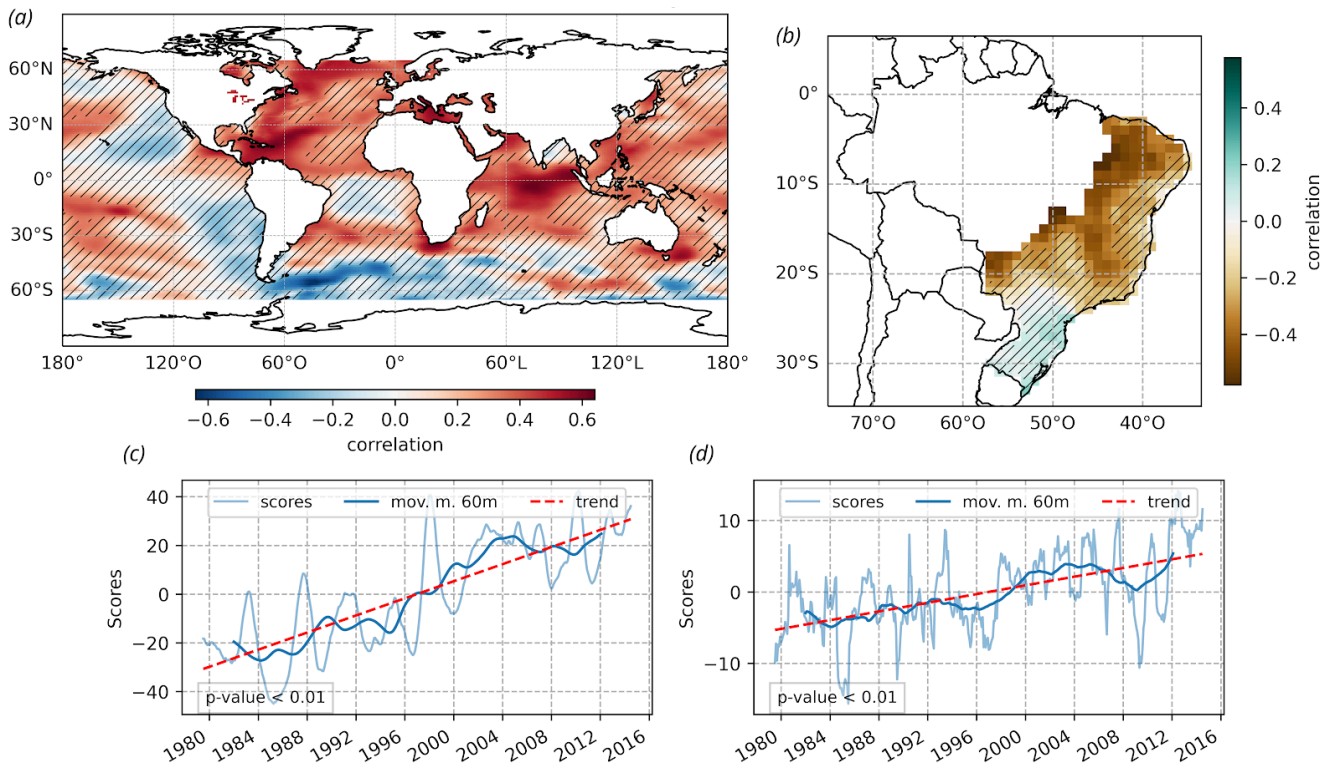

157

**Figure 2 - Second component of the PLS methodology applied using monthly precipitation data from ERA5 and SST data from COBE between 1979 and 2014. (a, b) The spatial maps represent the loadings matrices, where the hatchings represent areas where the statistical confidence on the sign of the anomaly is lower than 95%. (c, d) The time series represents the scores and the p-values indicate the statistical significance of the results.**

For the third and fourth components (Fig. S1 and S2), we observe more divergence in the results. Despite this, we consider these components in our evaluation due to their assigned weights and the potential for providing additional representation of driving mechanisms beyond the first and second components. Moreover, these third and fourth components might reveal important patterns that affect precipitation in Brazil, especially the Atlantic SST variability (Hastenrath and Greischar, 1993; Yoon and Zeng, 2010; Perez et al, 2022).

Through the analysis of the PLS components in the present climate datasets, we are able to select and rank the models based on their performance to reproduce these components. The model selection is based on a threshold of NRMSE< 0.6, and the individual model weights are based on the inverse of the average NRMSE among the PLS components scaled by the importance of each component, as described in the Methodology section. The table 2 lists the selected models and their respective weights along with the components these models skillfully represent, later employed to construct the weighted ensemble mean in the future climate section. These models selected through our approach are those that showed better performance in the task of simulating the impacts of precipitation in Brazil. This way, for example, the high weight of GFDL-ESM4, indicates that this model performs well in representing the overall components more accurately when compared to other models. While it is true that component 1 relates to the ENSO dynamics, the overall evaluation takes into

components that represent other important forcings of the Brazilian precipitation regime. For instance, the Atlantic SST variability drives the Brazilian precipitation variability in the Amazon (Yoon and Zeng, 2010), Northeast Brazil (Hastenrath and Greischar, 1992) and subtropical regions (Perez et al., 2022). Furthermore, by including multiple components in the analysis, we acknowledge that climate dynamics are multifaceted, and a comprehensive evaluation should account for more than just the primary modes of variability like ENSO. This approach rests on the importance of a holistic evaluation of model performance across various components, rather than focusing solely on the primary modes.

Table 2 - List of selected models and their weights represented as a percentage of their contribution to the ensemble mean.

| Model | Components | Weight (%) |
| --- | --- | --- |
| CAMS-CSM1-0 | 1, 2, 4 | 7.76 |
| CNRM-ESM2-1 | 1, 3, 4 | 7.73 |
| GFDL-ESM4 | 2, 4 | 7.59 |
| BCC-CSM2-MR | 1, 2, 4 | 7.37 |
| EC-Earth3-CC | 1, 2 | 7.11 |
| EC-Earth3-Veg-LR | 1, 2 | 7.08 |
| EC-Earth3-Veg | 2, 4 | 6.83 |
| IPSL-CM6A-LR | 2, 3, 4 | 6.69 |
| KACE-1-0-G | 1, 2 | 6.61 |
| CNRM-CM6-1-HR | 2, 3, 4 | 6.56 |
| MPI-ESM1-2-HR | 1, 4 | 6.28 |
| CMCC-CM2-SR5 | 1, 2 | 6.19 |
| FGOALS-f3-L | 2, 3 | 6.18 |
| MIROC6 | 1, 4 | 5.94 |
| CESM2-WACCM | 1, 4 | 4.08 |

**3.2 Precipitation mechanisms in the future climate (2020-2050)**

The oceanic mechanisms driving precipitation in Brazil in the future climate (2020-2050) are discovered by applying the PLS methodology in CMIP6 future climate simulations (Fig. 3 and 4). Figure 3 shows the first PLS component and Figure 4 the second PLS component; for each component, only models that performed well (NRMSE < 0.6) in the present climate are considered. The spatial maps show the average loadings matrices of the model ensemble, where each model is weighed by its skill in the present climate (Table 2); the hatched areas represent regions where at least 80% of the models disagree on the sign of the loadings matrix.

The first component shows a strong Niño-like pattern in the Central Pacific, similarly to what is found in the present climate
(Fig. 3a). However, unlike the present climate analysis, this Niño-like component shows a strong linear trend in the time
series of scores (Fig. 3c), suggesting that the climate models are mixing the natural variability of the ENSO phenomenon and
anthropogenic global warming; this warming trend can also be seen in the increasingly positive patterns in the tropical
Atlantic and Indian oceans. The impact of this warming trend in the Brazilian regional precipitation is a wetting pattern in
South Brazil and a drying pattern in Northeast Brazil, interfaced by a large region of uncertainty (Fig. 3b).
The second component illustrates a generalised warming trend in most regions of model agreement (Fig. 4a,c). This
component impacts precipitation in Brazil through a drying trend in the southernmost border of the country and a wetting
trend in the southeastern area. Some coastal areas in Northeast Brazil are significantly affected by a drying trend (Fig. 4b,d).
Although the linear trend was observed in most models (Fig. 3c,d and Fig. 4c,d), it becomes clearer and more robust in the
model subset; this reflects how sifting models in a mechanistic approach helps reduce the epistemic uncertainties.

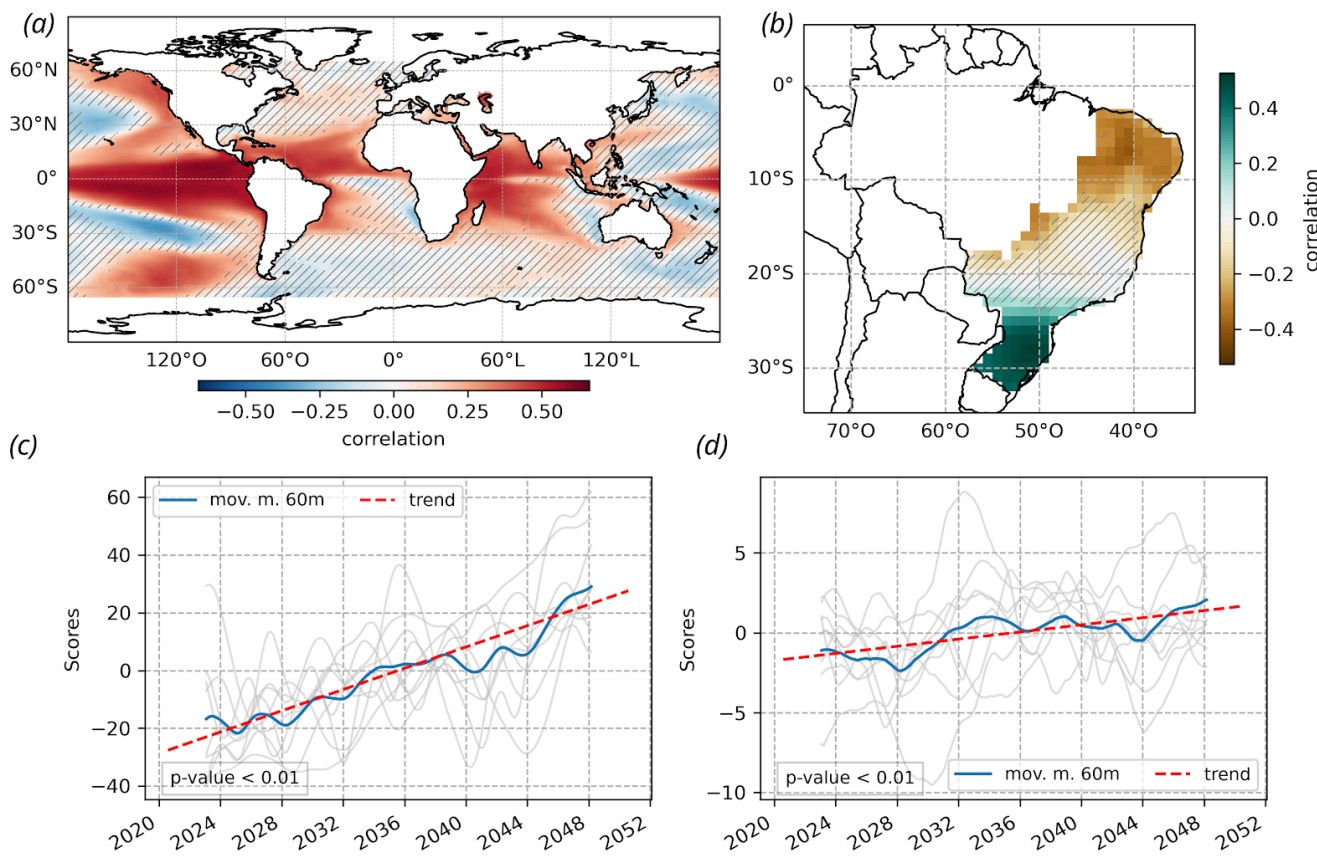

**Figure 3 - First component of the PLS methodology applied using monthly precipitation data from CMIP6 models under the**
**SSP2-4.5 scenario, listed in Table 2, between 2020 and 2050. The spatial maps represent the loadings matrices and the time series**
**represent the scores. The regions with hatching indicate areas of uncertainty with < 80% agreement in the sign change among the**
**models. The p-values indicate the statistical significance of the results.**

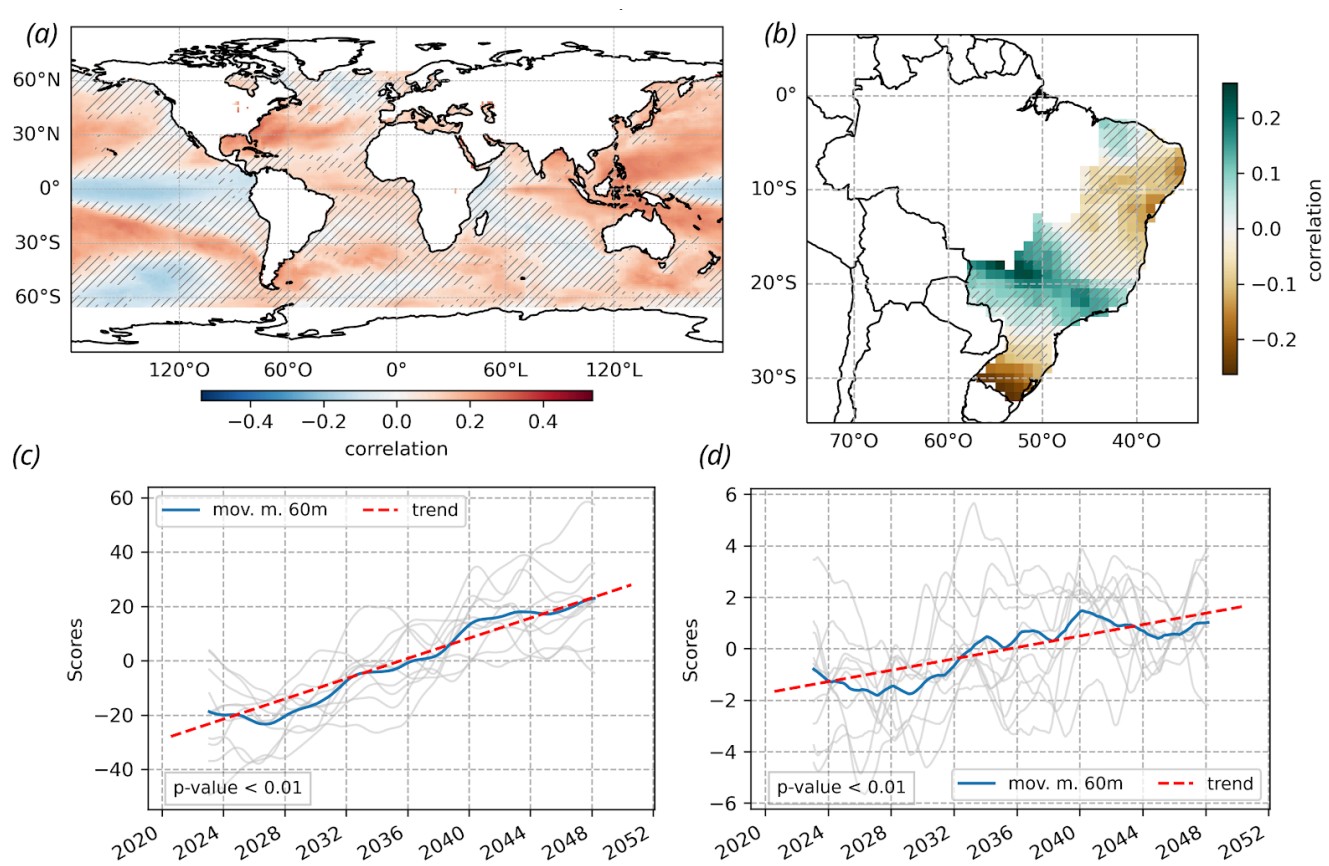


**Figure 4 - Second component of the PLS methodology applied using monthly precipitation data from CMIP6 models under the SSP2-4.5 scenario, listed in Table 2, between 2020 and 2050. The spatial maps represent the loadings matrices and the time series represent the scores. The regions with hatching indicate areas of uncertainty with < 80% agreement in the sign change among the models. The p-values indicate the statistical significance of the results.**

### 3.3 Future climate precipitation changes and uncertainty reduction

While the analysis of individual PLS components may support storyline approaches and mechanistic understanding, a quantitative precipitation change map is often required by decision-making bodies. With that in mind, we provide an uncertainty map based on the methodology employed by the IPCC in its 6th Assessment Report (Fig. 5). Here, we focus on the percentage of projected changes in 2020-2050 relative to 1979-2014. The hatching highlights regions where there is a significant lack of consensus, with at least 80% of the models analysed showing non-concordance, similar to the PLS uncertainty maps shown in the previous section.

Figure 5a shows the ensemble mean of the future precipitation changes using all CMIP6 models, listed in Table 1, while Fig. 5b uses the mean of the subset of models in Table 2 weighted by their skill in simulating precipitation mechanisms in the present climate (Fig. 5b). Firstly, we notice that the reduction of epistemic uncertainty by the proposed methodology is revealed by stronger anomalies and fewer hatched areas. This is further supported by the computed standard deviation

values, which show a reduction from 0.24 for the full ensemble set to 0.21 for the reduced subset—a decrease of 12.5%.
Particularly, the South Atlantic Subtropical High (SASH) shows stronger negative anomalies, suggesting a trend towards
drier conditions in the region via an intensification of the Hadley cell descending branch. Moreover, the positive changes in
South Brazil have increased after the application of the methodology; this enhanced dipole between the SASH and South
Brazil is consistent with the mechanism of restriction of cold fronts revealed by the PLS in the present climate and discussed
in Sect. 3a. In other words, selecting and weighting models that reproduce important precipitation mechanisms in the present
climate has increased the clarity of what may happen in the region in the near-future climate.

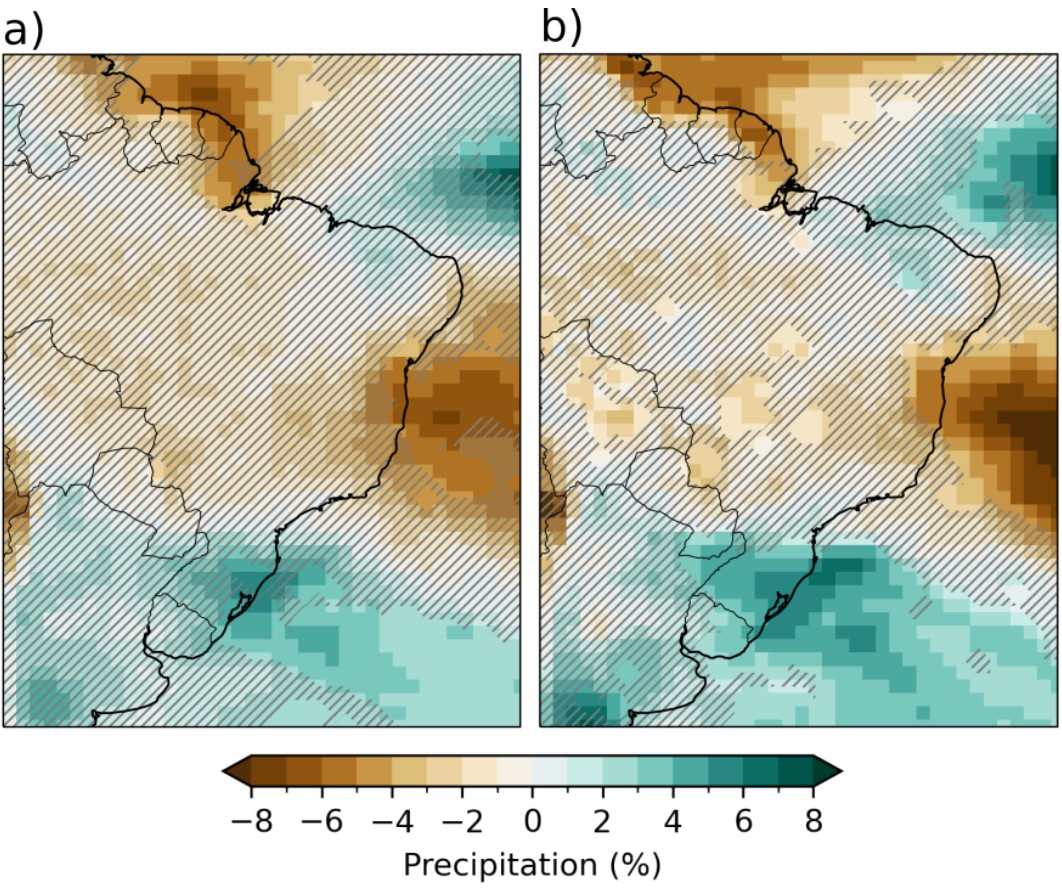

**Figure 5 - Percentual precipitation changes in 2020-2050 relative to 1979-2014 based on all assessed models, as listed in Table 1, (a)**
**and the percentual changes based on the selected models listed in Table 2 (b) from CMIP6 under the SSP2-4.5 scenario. The**
**regions with hatching indicate areas of uncertainty with < 80% agreement in the sign change among the models.**
Figure 6 shows the future precipitation changes broken down by season based on all models listed in Table 1 and only using
the models selected by the methodology (Table 2). A noticeable reduction of uncertainty across all seasons is evident when
comparing the hatched areas using all models versus only using the selected models, underscoring the success of our
process-based model selection methodology in enhancing our confidence in regional climate projections. The period from
December to May corresponds to the rainy season, characterised by a prevalence of uncertainties; this is in agreement with
Bazzanela et al. (2023) and Firpo et al. (2022), that also indicate that CMIP6 models perform better in the dry season than in
the wet season.
From June to November the Central and Northeast regions exhibit a clear drying pattern. In JJA, in particular, precipitation
in most of Brazil is largely driven by cold fronts, which, as previously discussed, can be restrained in higher latitudes if the
SASH is intensified. In SON, we expect an intensified SASH to also contribute to a later onset of the rainy season. This
drying pattern in JJA and SON is intensified in the subset of selected models. This is unsurprising, since the SASH
subsidence associated with an intensification of the Hadley circulation is one of the mechanisms discovered by the PLS
analysis in the present climate and used to select the best performing models.

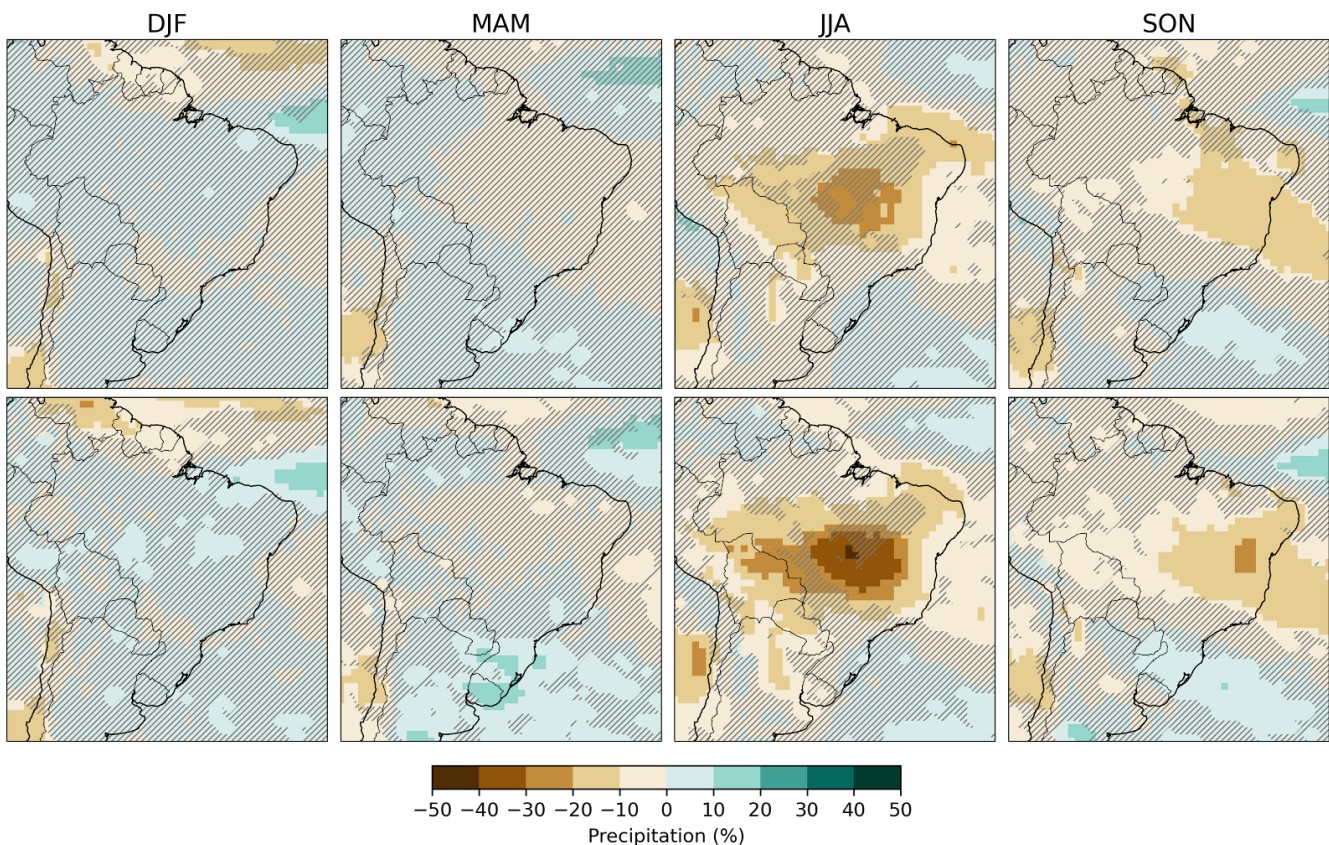

**Figure 6 - Seasonal percentual precipitation changes in 2020-2050 relative to 1979-2014 based on all assessed models, as listed in**
**Table 1, (up) and the percentual changes based on the selected models listed in Table 2 (down) from CMIP6 under the SSP2-4.5**
**scenario. The regions with hatching indicate areas of uncertainty with < 80% agreement in the sign change among the models.**

## 4 Summary and Conclusions

This study aims to reduce the epistemic uncertainty of regional precipitation changes in Brazil through a data-driven
process-based methodology of model selection and weighting. To achieve this, we first employ the methodology to discover
the main precipitation drivers in the present climate (1979-2014) in a reanalysis dataset (Sect. 3a), revealing that the El Niño
and the generalised warming of the oceans are linked to significant precipitation impacts in Brazil (Fig. 1 and 2). A distinct
positive linear trend in the global warming component is linked to a drying of most of Northeast and Southeast Brazil. We
propose that the linking mechanism between these SST and precipitation patterns is the intensification of the Hadley
circulation (Hu and Fu, 2007)  and, consequently, of the subsidence at the South Atlantic Subtropical High (Carvalho et al.,
260 2011).

The same methodology is then applied to CMIP6 present-climate simulations (Table 1) to evaluate the capability of CMIP6
models to simulate these precipitation drivers, thus creating a process-based model selection and weighting approach to
underpin the future climate analysis. From a total of 30 models, we select 15 models that are capable of simulating at least
two (Table 2) of the main regional precipitation drivers.
The mechanism discovery methodology is then applied to the near-future (2020-2050) climate simulations of the selected
models. We find that an ENSO-like pattern, tied to a generalised warming of the tropical oceans, is linked to an increase of
precipitation in South Brazil and a decrease in Northeast Brazil (Fig. 3 and 4), consistently with the present-climate
indication of an intensification of the Hadley circulation. This mechanistic view of regional precipitation changes can
underpin the development of storylines in future studies to support decision-making bodies in the water-energy-food nexus.
We go further to provide a quantitative view of regional precipitation changes based on the IPCC WG1 approach, contrasting
the uncertainty of precipitation changes using 30 CMIP6 models versus using the 15 selected models. We show that the
approach increased model agreement, particularly in South Brazil and SASH region. In the next 30 years (Fig. 6), a
noticeable reduction in uncertainty across all seasons is evident mostly from June to November. This period is characterised
by a clear drying pattern due to the strengthening of SASH, intensified within the subset of selected models, which leads to a
suppression of precipitation in Northeast and Southeast Brazil, possibly delaying the rainy season in these regions.
Our methodology of model selection and weighting considers the precipitation drivers rather than simply comparing CMIP6
model precipitation with observations. By selecting and weighting models mechanistically, we achieve a reduction of the
epistemic uncertainty of precipitation changes in Brazil in the CMIP6 ensemble. The method is based on the discovery of
statistical relationships between SST patterns and precipitation through the PLS and the assumption that models with an
accurate representation of these statistical relationships have a better representation of atmospheric processes leading to
precipitation. Considering that the atmospheric flow is the medium connecting SST and precipitation and the statistical
significance of the PLS loadings, we believe this assumption to be robust. However, as with any data-driven methodology,
there could be instances where confounding factors may influence the results; this highlights the need of other mechanistic
approaches capable of isolating rainfall mechanisms individually, such as atmospheric rivers, convergence zones, hurricanes
and fronts (Catto et al., 2015; Franco-Diaz et al., 2019; Perez et al., 2024).

**Competing interests**
The authors declare that they have no conflict of interest.

## Acknowledgments

This research results from the R&D project developed by MeteoIA for Engie Brazil, funded by ANEEL under Project No. PD-00403-0054/2022.

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
