# Peer review of "Data-driven discovery of mechanisms underlying present and"

_EGUsphere, 2024_

## Author Response (AR1)

**Author's Response**

**Data-driven discovery of mechanisms underlying present and near-future precipitation changes and variability in Brazil**
* * *
**Comments by Peter Pfleiderer**

The authors present an interesting model constraining application for precipitation changes in Brazil. The analysis is based on Partial Least Squares (PSL) regression. The scope and results of the study are highly relevant on a methodological as well as on a practical level. In the current state, the manuscript lacks some information on the method as well as some details in the results to allow a reasonable interpretation of the results. I would suggest major revisions before publication.

Although the method section is well written there are a few points that were not fully clear or could be misinterpreted due to lacking details:

1) How is the NRMSE calculated? You say that it is obtained by "comparing PLS scores and loadings between each model and those derived from the ERA5". How do you aggregate the comparison of scores and loadings? Do you weight score differences more than loading differences since the loadings have more features?

We thank the reviewer for this comment. The NRMSE was calculated between the loadings and scores from the models and the ERA5. For example, the X score of model A had its correlation calculated with the X score of ERA5, and so on. That means, we calculated scores and loadings separately and after that, we used RMSE normalisation. It is not the case that loadings have more features, because the lat-lon points were interpreted as samples, so correlation here is a metric of similarity between the loadings maps; it is important to recall that loadings do not have the time dimension. Scores, on the other hand, only have the time dimension, so correlation measures the similarity of the time series between pairs of scores.

We realise that this may cause confusion, so we expanded the explanation in the revised manuscript.

2) How many iterations of the PLS are done and how much of the variance is explained by the n(?) components? Is there a way to compute the variance explained by n components (similar to PCA)? For instance, is component 1 considerably more important than component 2?

Only the first four modes of the PLS were performed, but additional iterations could have been applied. It is possible to compute the explained variance by reconstructing the full dataset based on the scores and comparing it to the original dataset. Although we appreciate the question, this is without the scope of analysis of this work because the goal of the PLS is to retain the maximum co-variance between the two datasets rather than explaining the variance of only one dataset.

3) Is the first component in climate models always related to ENSO as shown in fig. 1 for ERA5 or are there some models where the first component resembles more a pattern as in fig. 2? If that would be the case and if component 1 and 2 would be similarly important, do you consider this when computing the NRMSE or do you always compare component 1 in the model with component 1 in ERA5? It would be interesting to see figures comparable to fig. 3 and fig. 4 for individual climate models.

In most models, the first component corresponds to the ENSO pattern. Due to the PLS method, it was decided to use the same component for each comparison, not mixing patterns from component 1 and component 2. We thank the reviewer for the suggestion and will be including a Supplementary Material with the individual figures for the first two components of each climate model.

My main question concerns the way how PLS is used to weight models: Can we assume that comparing individual scores and loadings identified in ERA5 and a climate model tells us how well the climate model reproduces the dynamics? Couldn't there be cases where different aspects/features of the SST forcing on precipitation (e.g. ENSO) are reproduced by the climate model but where the association of these aspects/features to components in PLS ends up to differ from ERA5? It would be helpful if the authors could discuss the assumptions made for the evaluation of model performance using PLS in more detail.

Thanks for raising that question, we have improved the discussion in the revised manuscript of the underlying assumptions and its possible shortcomings. We believe that it is true that models with dynamics more similar to reality will result in PLS scores and loadings closer to the reanalysis. This belief is based on the fact that the PLS seeks for relationships between SST patterns and precipitation patterns and that these relationships are physically mediated by the atmospheric dynamics; therefore, models with a more accurate representation of atmospheric dynamics should yield PLS components closer to the reanalysis.
However, as with any data driven methodology, there could be instances where there are confounding factors influencing our interpretation. To mitigate that, we will provide a clearer and more complete discussion of these pitfalls.

What are components 3 and 4? is there an interpretation for these components?
Thank you for pointing this out. We decided not to discuss these components for conciseness. We have included in the Annex components 3 and 4 for ERA5.

Table 2: How would you interpret the high weight of GFDL-ESM4? How skillful can a model be if it does not capture the ENSO dynamic (assuming that component 1 always represents some natural variability related to ENSO)? Or put differently, wouldn't we trust EC-Earth3-CC more as it robustly captures the ENSO and the climate change component? These points relate to my questions above concerning the comparability of components between models and ERA5.
We thank the reviewer for raising this point. Firstly, it is useful to remind that the goal of the method is not to select models that better represent the climate system as a

whole, but rather the ones that perform better at the task of simulating the impacts on Brazilian rainfall.

The high weight of GFDL-ESM4 indicates that this model performs well in representing the overall components more accurately when compared to other models. While it is true that component 1 relates to the ENSO dynamics, the overall evaluation takes into components that represent other important forcings of the Brazilian precipitation regime. For example, the Atlantic SST variability drives the Brazilian precipitation variability in the Amazon (Yoon and Zeng, 2010), Northeast Brazil (Hastenrath and Greischar, 1992) and subtropical regions (Perez et al., 2022). Furthermore, by including multiple components in the analysis, we acknowledge that climate dynamics are multifaceted, and a comprehensive evaluation should account for more than just the primary modes of variability like ENSO. Thus, GFDL-ESM4's performance signifies its robustness in capturing these diverse aspects effectively. This approach rests on the importance of a holistic evaluation of model performance across various components, rather than focusing solely on the primary modes.

Yoon, Jin-Ho, and Ning Zeng. "An Atlantic influence on Amazon rainfall." Climate dynamics 34 (2010): 249-264.

Perez, Gabriel MP, et al. "Using a synoptic-scale mixing diagnostic to explain global precipitation variability from weekly to interannual time scales." Journal of Climate 35.24 (2022): 8225-8243.

Hastenrath, Stefan, and Lawrence Greischar. "Circulation mechanisms related to northeast Brazil rainfall anomalies." Journal of Geophysical Research: Atmospheres 98.D3 (1993): 5093-5102.

Furthermore, I would find it interesting to have the NRMSE listed in the table. I would also find it interesting to see the NRMSE for models that are dropped due to lower skill.

This is a very interesting suggestion, we will include a table in the Annex with the requested information.

l128-129: Is the trend statistically significant? What do you mean by "scores do not show a strong linear trend"? I would agree, that trends in comparison to the trends of component 2, these trends are "weaker" but I still see a trend in fig. 1c.

In the revised manuscript, we have included a p-value testing the significance of the trends in the caption of each figure. We used the word 'strong' to highlight the difference with the second component. In the revised manuscript we reformulated this phrase to be clearer.

Fig. 5: Are the maps (a & b) ensemble medians? Or is it a mean considering the weights from table 2?

The Figure 5a shows an ensemble median, meanwhile the Figure 5b shows the mean between the models considering the weights from Table 2. Thanks for pointing this out, we have clarified in the revised manuscript.

Minor comments:

l56: does the "t" in "XtY" stand for transpose?
Yes, you are correct! We had an issue with file formatting regarding texts that were subscripted or superscripted.

equation 1: why is it max||u|| = ||v|| ?
Maybe there is a misunderstanding because of the formatting. In l61 we have:

$$Cov(\xi_1, \omega_1) \ = \ max_{\|u\|=\|v\|=1} Cov(Xu, Yv) \ .$$

l63-64: Check sentence. Is there something missing?
The correct is "the first pair of loading matrices is the one in which the corresponding latent vectors $\xi$ and $\omega$ are the most correlated." Thanks for pointing this out.

l162-163: Is this strong linear trend seen in most models or could it be that the trend is mostly due to a subset of climate models (relating to question 3).
Although the linear trend was observed in most models, it becomes clearer and more robust in the model subset; this directly reflects how sifting models in a mechanistic approach helps reducing the epistemic uncertainties.

l222-223: Do you have a reference supporting this hypothesis?
A number of studies point the intensification and poleward expansion of the Hadley circulation caused by Global Warming (e.g., Hu and Fu (2007)). The South Atlantic Subtropical High is the descending branch of the Hadley Circulation, therefore, its poleward movement and intensification affects precipitation in South America (Carvalho et al., 2011).
Hu, Y., and Qinjun Fu. "Observed poleward expansion of the Hadley circulation since 1979." Atmospheric Chemistry and Physics 7.19 (2007): 5229-5236.
Carvalho LMV, Jones C, Silva AE, Liebmann B, Silva Dias PL. 2011. The South American Monsoon System and the 1970s climate transition. Int. J. Climatol. 31: 1248–1256, doi: 10.1002/joc.2147.
* * *
**Comments by Elena Saggioro**

*Introduction*:

There is a lack of reference to previous analysis of the link between SST and precipitation in the region (only mentioned in L45). Can you please provide a brief overview, to better locate the contribution of this study?

Thanks for the suggestion. We have included more references discussing the links between SST and precipitation in the region, such as Grimm et al. (2000) and Coelho et al. (2002).

Grimm, Alice M., Vicente R. Barros, and Moira E. Doyle. "Climate variability in southern South America associated with El Niño and La Niña events." *Journal of climate* 13.1 (2000): 35-58.

Coelho, Caio A. dos S., Cintia Bertacchi Uvo, and Tércio Ambrizzi. "Exploring the impacts of the tropical Pacific SST on the precipitation patterns over South America during ENSO periods." *Theoretical and applied climatology* 71 (2002): 185-197.

*Method:*

I would appreciate a more detailed introduction to the PLS method:

Can you rephrase in physical terms what "maximize the information present in XtY" mean? (Is it the correlation in time between SST and Precip at two different locations?)

We used the expression "maximize the information" as a concise way to convey that the PLS method finds latent variables that represent the maximum covariance between the SST and precipitation. This is done considering all the locations at the same time, rather than the correlation between SST and precip at two different locations as suggested by the reviewer. We have clarified this in the revised manuscript.

1- Could the authors expand on how the modes are identified (e.g. Where can we see the "modes" from Eq 1? )

In Eq 1, the modes can be seen as the pairs of scores $\xi$ and $\omega$ and pairs of loadings X and Y. In the originally submitted manuscript, there was a formatting problem, the equation was meant to be as follows:

$$Cov(\xi_1, \omega_1) = max_{\|u\|=\|v\|=1} Cov(Xu, Yv) \, .$$

2 - Can you give, as example, how the reader should interpret two "loading patterns" in relation to each other (e.g. for mode 1 in Fig1.a and Fig1.b)?

Answer: The loading patterns are connected via the scores timeseries. For example, if the SST sign is negative at a certain location and the score sign is positive, it means that it is negatively associated with the corresponding map of precipitation loadings. We have included this expanded explanation in the revised manuscript.

3 -I would find helpful if the authors could clearly define each term (e.g. mode, loading, scores), associate with a mathematical symbol and show their formula where relevant. Please then repeat the symbol each time it is mentioned in the Methods section, to help the reader connect the terms/formula more easily. Also, as

noted in the technical corrections, the use of this terminology is at times inconsistent in the text/figures captions.

We encountered an issue with file formatting regarding subscripted or superscripted texts, and some mathematical symbols were deleted during the article formatting, which has already been promptly corrected in the revised manuscript. We have also corrected the inconsistencies and included the mathematical symbols where relevant.

How do you combine the NRMSE from the scores and loads into one value (for each mode)? (L93)

The RMSE is initially calculated for each pair of loadings (models versus ERA5) and scores (models versus ERA5). Then, we normalise the RMSE between 0 and 1 to obtain the NRMSE. Finally, we multiply (1-NRMSE) by the weights in order to obtain a single rank value where higher values are better.

The weights reflect the importance of each mode through the coefficient of determination in order to reflect how well does a single score represents the original precipitation data, thus favouring "more relevant" modes in the ranking procedure.

This has been described in more detail in the revised manuscript.

*Present climate results:*

To increase the readers' trust of the selected models, it would be good to see the 1 and 2 components of the models to get a feeling of how well they perform compared to "observation" beyond the NRMSE. A selection could appear in the Supplementary Material and only referenced in the text.

Thank you for your suggestion. We have provided the suggested figures in the Supplementary Material.

What do components 3 and 4 represent? Why using them, in case they are not linked to any physical mechanisms?

The ERA5 components can identify certain subtle patterns, and when comparing them to the models, we observe more divergence in the results, particularly with components 3 and 4. Despite this, we consider these components due to their assigned weights and the potential for providing additional representation of driving mechanisms beyond components 1 and 2. Moreover, components 3 and 4 might reveal important patterns that affect precipitation in Brazil, especially the Atlantic SST variability (Hastenrath and Greischar, 1993; Yoon and Zeng, 2010; Perez et al, 2022).

Yoon, Jin-Ho, and Ning Zeng. "An Atlantic influence on Amazon rainfall." *Climate dynamics* 34 (2010): 249-264.

Perez, Gabriel MP, et al. "Using a synoptic-scale mixing diagnostic to explain global precipitation variability from weekly to interannual time scales." *Journal of Climate* 35.24 (2022): 8225-8243.

Hastenrath, Stefan, and Lawrence Greischar. "Circulation mechanisms related to northeast Brazil rainfall anomalies." *Journal of Geophysical Research: Atmospheres* 98.D3 (1993): 5093-5102.

*Future climate results:*

What is the implication of models that do not represent well some of the first 4 components selected? (see Table 2; some models do not represent component 1 even which seems to be crucial). Does considering all 4 of them regardless not result in possibly selecting models that actually behave very differently?

Answer: Although components 1 and 2 (ENSO and global warming) are crucial, other modes of variability may drive precipitation in Brazil even more directly, such as the Atlantic SST variability mentioned in the previous answer. By including multiple components and weighting them by importance, we acknowledge the complexity and multifaceted nature of the climate system.

Yes, considering all 4 of them might result in selecting models that behave differently, but this is a desired feature of the analyses because we aim to represent the epistemic uncertainty associated with how different models represent reality.

To allow for clearer link between components and decrease in uncertainty in Fig 5.b, it would be interesting to see what changes to Fig5.b if:

Only the models that match ERA for at least Component 1 are included (e.g. no GFDL-ESM4 as seen from Table 2): will the Component 1 of the precipitation signal dominates the overall projected change from the models?

Only the models that match ERA for at least Component 1 and 2 are included (e.g. no *CNRM-ESM2-1* as seen from Table 2.

These tests are suggested because it seems that most of the drying in the north/wetting in the south is due to Components 1 and 2. Hence, I would imagine the models that represent them will be the ones that reduce the uncertainty and reveal that pattern.

Thank you for your insightful suggestion. I believe that selecting only models that perform well for Component 1 and Component 2 could underestimate the epistemic uncertainty among the models, and perhaps seem like cherry-picking. However, we appreciate the suggestion and have decided to test running the figures like the reviewer suggested. If it leads to substantial changes in Fig 5b, we will include it in the Supplementary Material.

Further, it would be interesting to see what happens to Fig5.b if no weighting is applied to the selected models (but just a simple average is taken): is the weighting very important, or does the PLS method identified models are already "better" without the need for weighting?

This question is very interesting! Although not by a large amount, weighting the selected models lead to noticeably less uncertainty hatchings and improving the results overall.

*Discussion:*

While I do not think the following is the case, it still would be good for the authors to comment on how the reduction in the uncertainty in precipitation changes for the PLS ensemble versus the full one is not an "automatic" result deriving from the construction of the procedure itself. I think this is not the case, because the selection is done on the past climate, not on the future. But it would be good to elaborate on this as it is a question that often arises for filtering methods like this one.

Thanks for the comment. We also believe that this is not the case, since the procedure was "trained" on past climate data. We will expand on that in the revised manuscript.

Finally, I would suggest adding a comment on the assumptions of this method. If I understand correctly, this approach rests of the assumption that the features detected in the past for the CMIP6 models (via the PLS procedure) are going to identify models that will also behave closer to reality in the future. More specifically, it seems that:

1. the physical assumptions are in the future, some of the dominant features relevant to Brazil precipitation will be linked to SSTs (and more specifically to ENSO and generalised warming of the oceans) that the models that better represent these connection in the past will continue to be the most able to represent it under increasing forcing in the future while the methodological assumption is that the PLS method can reliably detect the models with the correct mechanisms representing the physical connection between SST/ENSO and precipitation.

These are justifiable assumptions, but I think a discussion of them and their limitations are missing.

We thank the reviewer for this suggestion. This was also suggested by Reviewer 1. An expanded discussion of the assumptions and potential limitations will be included in the revised manuscript.

Technical corrections

Data: are anomalies or full field used?

We employed anomalies in order to filter out the seasonal variability.

l45: is this a separate question (then better to phrase with question mark for consistency with the style of the first question) or a comment to the first question above?

It was a comment to the first question. Thanks for noticing.

L49: this question is phrased rather oddly, a rewrite would be useful. ( adding also a question mark at the end for consistency with the style of the first question)

We have rephrased this question as ''What are the predictions for precipitation in Brazil over the next 30 years based on a mechanistic filtering and weighting procedure?''

L54: introduce lat-lon as "latitude – longitude (lat-lon)"
Thanks for your point. We have included this.

L56: Xt means transpose?
Yes. Thanks for noticing, we have clarified in the text. We had an issue with file formatting regarding texts that were subscripted or superscripted.

L56: seems there is an extra "and"? in "The PLS method identifies pairs of latent variable vectors and that maximises…."
Yes, we have corrected in the revised version as 'The PLS method identifies pairs of latent variable vectors $\xi$ and $\omega$ that maximises the information present in $X^tY$.'

L59-62: Please explain in simple terms why you get the modes from this equation. Does "Xu" stands for matrix-vector product between X and u? What is the dimension of u?
Due to malformatting, the equation was displayed incorrectly in the original manuscript. Eq 1 should be read as:

$$Cov(\xi_1, \omega_1) \; = \; max_{\|u\|=\|v\|=1} \; Cov(Xu, \, Yv)$$

Where X represents the SST, u represents the SST loadings, Y represents precip and v the precip loadings. The matrix-vector product between X and U yields the scores vector $\xi_1$ and the product between Y and v yields $\omega_1$.

L79: can you comment briefly on why was ERA5 chosen instead of GPCP directly for precipitation?
GPCP data have a shorter time period (from 1979) when compared to ERA5 data, which is available since 1950. Since we aimed to investigate interannual/interdecadal variability and climate change, we chose the longer dataset.

L85: can you comment briefly on why SST used from COBE and not ERA5?
Because COBE SST also presents a long temporal availability and is observations-based.

L108-109: Why does the ratio between 2020-2050 and historical ensemble mean climatology is interpreted as "uncertainty?" Is this not the climate change signal, instead? And do you maybe mean a change (future-past) is at the numerator?
Yes, the word uncertainty was misplaced there. The intent was to explain how the mean change is computed. This has been fixed now.

L119: is the Amazon region removed before or after the PLS analysis?

The Amazon region was removed before the PLS analysis so that the magnitude of precip variability there would not dominate the subsequent analyses. We have made this clearer in the revised manuscript.

L120: can you comment on why "the precipitation in the Amazon region presents significantly higher variability" : is this because of the precipitation induced by transpiration from the trees? Is this choice of cropping out the Amazon forest something done in other papers too?

Precipitation in the Amazon is much higher in magnitude than other regions in the country; it varies almost directly depending on ENSO, tropical Atlantic SSTs and evapotranspiration from trees. Many studies using PCA showed patterns where the Amazon region dominated the analyses. Since this paper was focused on regions more socioeconomially active, we decided to crop out the region to have clearer results in the area of interest.

L139: was it not between 1979 and 2014? (see L85)

Thanks for noticing that. It was between 1979-2014, not 1979-2015. We corrected that.

L140: what is the anomaly? I have noticed a somewhat inconsistent (or unclear) use of the terms "saliences – correlation" (figure title), "anomaly" and "loading matrices" (caption) : please clarify.

We have fixed the manuscript for consistency. The maps show the loading matrices, which correspond to the saliences of PLS analyses computed as correlations. So the terms are somewhat interchangeable.

L199: is the map showing a ratio between the CHANGE in precipitation and the past [Precip(future)- Precip(Past)]/ Precip(Past), or a ratio between the values Precip(future)/Precip(Past)? It is not clear from the text or the caption.

The ratio is between the values [Precip(future)- Precip(Past)]/ Precip(Past). We reformulated this point to be clearer.

Fig 1.c: There is a negative trend in Figs 1.c (not commented on, only for Fig 1.d): could you elaborate on it? Is this linked to any observed trends in variability in the region? How does this relate to the interpretation of Lines 66-69?

There is a negative trend in Fig. 1c scores, indicating a change of sign in the patterns of Fig. 1a. This suggests that, in the first half of the timeseries, El Niño conditions were dominant, while in the second half of the time series, La Niña conditions were dominant. This relates to the explanation of lines 66-69, that describes how the relationship between loadings and scores should be interpreted.

Fig 5,6 : I would find more intuitive and consistent to use the brown-green colorscale here since we talk about precipitation change? Why not cropping the Amazon are from here too?

Thanks for your suggestion. We decided to use the red-blue colorbar as it is universally used to represent positive-negative dichotomies. Moreover, the suggested colorbar has already been employed in Figures 1-4, so we believe a different colorbar could enhance reader comprehension. The Amazon was excluded from the PLS analyses as to not dominate the signal. Once the selecting and weighting is performed, there is no reason to crop out the Amazon.

---

## Author Response (AR2)

Public justification (visible to the public if the article is accepted and published):
The revised manuscript has been seen again by one reviewer, which suggest publication after addressing some further revisions.

Please make sure that all captions are complete and comprehensively document all panels and lines shown in them in detail.

Line 35 Are all models for the experiments used here? It may make sense to only count models that also provide output for the experiments analyzed here.
Thank you for pointing this out. We have included a complementary phase.

Line 44: Missing blank space after question mark.
Thank you for noticing. We have corrected the revised version.

Line 48: I suggest using the term "projections" in "predictions" as in IPCC to refer to scenario-based projections as opposed to "predictions with initialized SST conditions".
We have made corrections in the revised version.

Line 75: Specify whether the radiative forcing and land use changes are also changing or whether they are kept constant.
The corrections have been included in the revised version.

Line 100-101: The sentence reads odd and it is unclear what is normalized. In the current formulation it is unclear whether the error first normalized at the grid scale level or if it is the total RMSE normalized in the end?
After calculating the RMSE for each model, these values were collectively normalized to ensure consistent scaling across all comparisons, we have clarified in the text.

Line 148-152: Please refer to panels a-d in the figure caption and provide a more comprehensive documentation on what is shown in each individual panel.
Thank you for pointing this out. We have corrected in the revised version.

Line 153-156: Likewise, refer
We have corrected in the revised version, too.

Line 162-163: Is there any AMIP run with multiple realizations in which you could test how robust the PLS loadings are when running multiple members of AMIP within the exact same model but with different atmospheric initial conditions.
Thank you for the thoughtful suggestion. Testing the robustness of the PLS loadings across multiple realizations within the same model, with different atmospheric initial conditions, is indeed an interesting approach. However, this falls outside the scope of the current project that was funded between 2022 and 2023. In the conclusion section, we have included a sentence about this limitation and highlighted the opportunity for future studies.

Additionally, we expect that aleatoric variations from different initialisations will not affect the large-scale patterns identified by the PLS methodology. Similarly to Principal Component Analysis, PLS isolates dominant and clear patterns of variability in datasets. While the noise introduced by the different initialisations will impact the weather patterns and subsequently the temporality of climate oceanic patterns, the relationships between these patterns and precipitation in South America is not expected to change, since these are due to the model's ability in representing the atmospheric physics and its relations with the oceans, which should be kept equal in a single-model ensemble.

Line 217-218: Please add a comparison of a measure of ensemble spread to support the reduction in uncertainty between full ensemble and subset
Thanks for your point. We have now computed the standard deviation of the full ensemble set (0.24) and the reduced ensemble set (0.21), meaning a reduction of 12.5%.

Additional private note (visible to authors and reviewers only):
Note that one reviewer was no longer available to review the revised manuscript. Therefore, I added some additional points above. Please carefully address the questions by the reviewer and myself regarding the robustness of the results and the sensitivity to the chose threshold 0.6 and the realization of the climate model.
We have furthered the discussion on the 0.6 threshold in the methodology section.

---

## Author Response (AR3)

The reviewers have answered all my comments and clarified parts of the methods section which makes it more readable and understandable. The additional material in the supplementary information also helps the interpretation of the analysis. Overall, the manuscript is in a good shape. While reading the manuscript again (with a better understanding of the method) I got the following question (apologies for not having asked this in the previous round):

Do we expect the scores of the SST component 1 (ENSO) pattern from ERA5 to be correlated with free running climate simulations? For component 2, the authors argue that it represents mainly a forced warming signal and therefore should be similar in ERA5 and model runs (l149). I see how for component 2 this could be meaningful. For component 1, the authors write "This suggests that in the first half of the timeseries, El Niño conditions were dominant, while in the second half of the time series, La Niña conditions were." (l144). I would argue that this reflects natural variability. Therefore, I'm wondering why the correlation of the scores of component 1 between ERA5 should be correlated to the corresponding scores of a freely running climate model simulation. Does the correlation of scores really inform about the skill of that climate model to represent the dynamics. Wouldn't it be more meaningful to calculate the NRMSE score only for the loadings (ignoring the scores)?

Thanks for the insightful comment. Yes, we expect the scores of the simulations to match ERA5 because the simulations are not freely running; the simulations are from AMIP and therefore forced with the observed SST. Thus, it is a good sign if the model timeseries and reanalyses timeseries are in agreement, even for natural variability modes; this suggests that the atmospheric model is skillful in translating these SST patterns into precipitation.

Further suggestion: The authors made a number of methodological choices (0.6 NRMSE threshold, threshold must be met by at least 2 components, considering the first 4 components, ...?). While I see the necessity of such methodological choices, I would appreciate a little sensitivity analysis where some of these choices are altered and the results from figure 6 are shown again. Alternatively, the choices could be listed and shortly discussed in the discussion section again.

Thank you for the suggestion; we agree it would be ideal to test the sensitivity of these choices. Unfortunately, this falls outside the scope of the current project, which was funded between 2022 and 2023. However, in the revised version, we did provide a brief discussion of these choices in lines 108-111 and again in the conclusion section (l. 288-290).

L57: "latents variables"
Thank you for pointing this out. We have corrected the revised version.

L73: "For instance, if the SST signal is negative at a particular location while the score is positive, this indicates a negative association with the corresponding precipitation loading pattern." Thanks for adding this explanation, i think it's useful. I think this example should be even more specific. What has a "negative association with the corresponding signal"? And how does the time dimension come in? This seems important to me as the scores change sign over time in your example. Would the following be correct? "For instance, if the SST loading is positive in the ElNino region while the score is negative in a given year, this indicates that in that year, warm SSTs in the El Nino region are negatively associated with the corresponding precipitation loading pattern."-

We thank the reviewer for the suggestion and improved the explanation between L.73 and L75.

L101: "were separately calculated and after that, used a normalisation." -> as normalization?
In the last revision we modified this section and believe it is now written in a clearer and more detailed way. With this, we believe that the use of the term 'normalization' is now properly contextualized and aligned with the intended meaning we want to convey.